# Dietary Heat-Treatment Contaminants Exposure and Cancer: A Case Study from Turkey

**DOI:** 10.3390/foods12122320

**Published:** 2023-06-09

**Authors:** Hilal Pekmezci, Burhan Basaran

**Affiliations:** 1Department of Elderly Care, Health Care Services Vocational School, Recep Tayyip Erdogan University, Rize 53100, Türkiye; 2Department of Nutrition and Dietetics, Faculty of Health, Recep Tayyip Erdogan University, Rize 53100, Türkiye; burhan.basaran@erdogan.edu.tr

**Keywords:** acrylamide, heterocyclic amine, polycyclic aromatic hydrocarbon, N-nitrosamines, meat, French fries, tea, bread, coffee, dietary habits, cancer

## Abstract

In this study, the 10-year dietary habits of patients diagnosed with cancer (*n* = 1155) were retrospectively analyzed, and the relationships between dietary (red meat, white meat, fish meat, French fries, bread, instant coffee, ready-to-drink coffee, Turkish coffee, and black tea) heterocyclic amines, polycyclic aromatic hydrocarbons, acrylamide, and N-nitrosamine-based risk scores and cancer types were statistically evaluated. The foods with the highest and lowest mean dietary heat-treatment contaminant risk scores were red meat and ready-to-drink coffee, respectively. There were statistically significant differences in the dietary heat-treatment contamination risk scores based on the cancer patients’ demographic characteristics (sex, age, smoking, and body mass index) (*p* < 0.05). According to the cancer types, the systems with the highest and lowest dietary heat-treatment contaminant risk scores were determined as other (brain, thyroid, lymphatic malignancies, skin, oro- and hypopharynx, and hematology) and the reproductive (breast, uterus, and ovary) system, respectively. The relationship between instant coffee consumption and respiratory system cancer types, the frequency of consumption of French fries and urinary system cancer types and the consumption of meat products and gastrointestinal system cancer types were determined. It is thought that this study contains important findings regarding the relationship between dietary habits and cancer and will be a good source for other studies to be conducted in this context.

## 1. Introduction

Cancer is a highly prevalent disease globally and has become a new normal, especially in the growing generation [1,2]. Cancer is the second leading cause of death after cardiovascular diseases [3]. Sung et al., (2021) estimated that 9.9 million of 19.3 million cancer cases resulted in death in 2020, and by 2040, cancer cases will increase by approximately 50% and reach 28.4 million [4]. It has been reported that modifiable environmental risk factors, such as unhealthy eating habits, smoking and alcohol consumption, excessive body weight, and physical inactivity, significantly contribute to cancer development [5,6].

Dietary patterns are risk factors for multiple types of cancer [7]. A healthy diet has been reported to reduce the risk of developing many types of cancer [8,9]. Chemical fertilizers used in the production of primary plant and animal foods, pesticides, additives added to foods during production, inappropriate storage conditions, and individual consumption habits cause the formation of toxic compounds in foods [10]. Poisonous examples of the toxic compounds in question are heterocyclic amines (HCAs), polycyclic aromatic hydrocarbons (PAHs), acrylamide, and N-nitrosamines, which are formed with heat-treatment applications and associated with cancer.

HCAs occur with prolonged heat treatment of various foods, especially protein-rich foods such as meat and fish, at 100 °C and above [11]. The formation pathways of HCAs in food have been explained by two mechanisms [12]. The first category comprises aminoimidazoazoarenes, also known as “thermic HCAs” or “IQ type HCAs”, which are produced at temperatures between 100 and 300 °C. The pyrolysis of amino acids and proteins at temperatures above 300 °C results in the formation of the second category (amino carbolines), also referred to as “pyrolytic HCAs” or “non-IQ type” [13,14]. More than 30 HCAs have been identified in cooked foods [15]. Most HCAs have been recognized for their potential cancer-causing or mutagenic effects on humans [16]. According to previous studies, HCAs are 100 times more potent than aflatoxin and 2000 times more mutagenic than benzo[a]pyrene [17]. Certain heterocyclic aromatic amines have been classified as probably carcinogens to humans (Group 2A) and possibly carcinogens to humans (Group 2B) by the International Agency for Research on Cancer (IARC) [18].

Nitrate and nitrite are additives added to specific food categories, especially meat products, to impart various functional properties. They are also present in foods because of natural occurrence (the nitrate–nitrite pathway) or contamination (from the soil, water, or air) [19]. Nitrate is not directly toxic, but bacteria in the oral flora and gastrointestinal tract reduce nitrate to nitrite, which can react with secondary amines and other nitrogenous substances to form compounds that threaten human health, such as nitric acid and N-nitrosamines [20,21]. N-nitrosamine compounds are also included in various foods, especially heat-treated meat products, and are taken into the human body with food consumption [22]. The c N-nitrosamine compounds’ carcinogenic, mutagenic, and teratogenic effects have been described, and their presence in food is recognized as a reasonable public health concern [23]. N-nitrosamine compounds have been identified in a fairly broad range in humans, including those that are directly carcinogenic (Group 1), probably carcinogenic to humans (Group 2A), possibly carcinogenic to humans (Group 2B), and unclassifiable as to carcinogenicity in humans (Group 3) [18].

Acrylamide is a colorless, odorless, and crystalline compound with a molecular weight of 71.08 g/mol, easily soluble in acetone, ethanol, methanol, and water [24]. Depending on the food type, the primary mechanism for acrylamide formation is the Maillard reaction which occurs at high levels with heat treatment above 120 °C, especially in carbohydrate-rich foods [25,26]. Many researchers and organizations have reported that acrylamide occurs at different levels in many foods commonly consumed daily, such as coffee, French fries, bread, biscuits, breakfast cereals, meat products, and various traditional foods [27,28,29,30,31]. Recent research has shown that acrylamide has carcinogenic, neurotoxic, genotoxic, immunotoxic, and mutagenic effects on animals [32]. Acrylamide was classified by the IARC in Group 2A as “probably carcinogenic” in 1994 [18]. In the European Union legislation on chemicals, acrylamide is a carcinogen and mutagen in Category 1B and a reproductive toxicant in Category 2 [33].

PAHs are ubiquitous environmental pollutants released in large quantities from many sources, such as coal processing, motor vehicle exhaust, forest fires, volcanic eruptions, etc., [34,35]. Therefore, the food chain can easily be contaminated by PAHs in the environment [36]. In addition, PAHs can be formed after processes such as smoking, heating, roasting, and drying are applied to foods [37]. Most PAHs are carcinogenic, teratogenic, and mutagenic [35,36]. PAHs may also exhibit synergistic properties [38]. Edible oils, meat and dairy products, cereals, confectionaries, coffee, and tea are essential sources of PAH contamination [39,40]. Therefore, the presence of PAHs in food is a concern [41]. Benzo[a]pyrene (BaP) is the most widely studied PAH [42]. The International Agency for Research on Cancer (IARC) has identified some PAHs in Group 1–2B (directly–possibly carcinogenic to humans) [18]. The United States Environmental Protection Agency has included PAHs in its list of priority organic pollutants [43].

The incidence of cancer is increasing worldwide [44]. Therefore, individuals need more information on issues that significantly improve their potential of developing cancer and are often related to their lifestyle [45,46]. The World Cancer Research Fund/American Institute for Cancer Research has evaluated evidence on dietary factors and various cancer risks for more than 20 years [5]. In the literature, the relationship between the dietary intake of heat-treatment contaminants, which are the focus of this study, and cancer has been examined in some studies [47,48,49,50,51,52,53]. A common feature of these studies is that they mostly contain evidence for a single food and heat-treatment contaminant. This situation falls short of covering the complexity of nutrition, including interactions between all foods and their components [54]. The foods individuals consume, how much they consume, how they prepare them, and how they consume them directly impact their health because dietary exposure continues throughout life [55]. This study considered meat, French fires, bread, coffee, and black tea risky for exposure, which are frequently consumed by individuals in daily life and are considered risky for the formation of HCAs, PAHs, acrylamide, and N-nitrosamines (some or all). This study focused on the consumption characteristics of the foods in question in cancer patients’ dietary habits relationship between the risk scores calculated for exposure to HCAs, PAHs, acrylamide, and N-nitrosamines based on the dietary habits of patients and cancer types that were statistically analyzed.

## 2. Material and Methods

### 2.1. Ethical Standards Disclosure

This study was conducted according to the “Principles of the Declaration of Helsinki”. With a decision letter dated 6 December 2019 and numbered 2019/179, the university’s ethical committee where the study was conducted granted written consent. In addition, permission dated 29 November 2019 and numbered 64247179-799 was received from the Provincial Health Directorate for institutional approval. Because the use of human data in the research requires the protection of individual rights, the “Informed Consent” condition and voluntariness, which are fulfilled as ethical principles, were taken as a basis. Moreover, the principle of “Respect for Human Dignity” was also taken into consideration in the study, and according to the “Confidentiality Principle”, the individuals who would be participating in the study were told that the information about them would not be disclosed to others.

### 2.2. Study Population

Between 1 January 2020 and 30 October 2022, all patients who visited an oncology center in Turkey, knew their cancer diagnosis and chemotherapy, were 18 years of age or older, had the ability to understand the statements in the questionnaire, did not have a diagnosis of a psychiatric disorder, had place, time, and person orientation, and voluntarily agreed to participate in the study constituted the study population.

### 2.3. Dietary Questionnaires

To determine the risk scores of exposure to heat-treatment contaminants arising from the dietary habits of patients diagnosed with cancer, the researchers prepared a questionnaire titled “Patient Information and Retrospective 10-Year Dietary Habits”, consisting of three sections, after reviewing the literature. In the first part of the questionnaire, the purpose of the study, explanations regarding the necessary ethical and administrative permission documents, and the researchers’ names, contact information, and institutional information were defined. The second part of the questionnaire included a total of nine questions about demographic characteristics such as gender, marital status, age, occupation, body weight, height, whether they smoked in the past, whether they had a family history of cancer, and the type of cancer they were diagnosed with. In the last section, questions were asked to shed light on the dietary habits of individuals over the past ten years. This study is the first to identify food as a potential risk for the occurrence of targeted heat-treatment contaminants. In this content, nine sub-food types in five different food groups were defined. This section divides each food’s consumption frequency, portion size, cooking method, and consumption style into sub-parameters. The determination of each variable and the risk scores of the variables are explained below.

#### 2.3.1. Selection of Food Groups

This study focused on HCAs, PAHs, acrylamide, and N-nitrosamine compounds. In this context, some foods frequently consumed by patients in their daily lives and considered risky in terms of the formation of these heat-treatment contaminants were included in the study.

Meat: Meat products are frequently consumed by society because of their high protein content and rich mineral and vitamin content. The formation of HCAs, PAHs, acrylamide, and N-nitrosamine compounds in meat products and the relationship between meat consumption and various types of cancers have been examined in many studies [56,57,58,59]. In this study, the meat group was divided into three subgroups: red meat (bovine and ovine red meat), white meat (poultry meat), and fish meat.

French fries: Potatoes are among the most widely produced and consumed foods worldwide [60]. Potatoes are mostly consumed as fried in oil [61]. HCAs, acrylamide, and PAHs have been reported in French fries [62,63,64]. Therefore, the relationship between French fries and cancer has been evaluated in many studies [65,66]. French fries were included in the study because of the high occurrence of acrylamide, their high consumption, and the toxicity of acrylamide.

Bread: Bread is a globally important food in terms of nutrition. It has been reported that HCAs, PAHs, and acrylamide are formed in bread, and the relationship between bread consumption and cancer has been examined in many studies [67,68,69,70,71,72]. Turkey ranks first in bread consumption worldwide [73]. Many types of bread are sold in Turkey. However, 85% of the population consume white bread [74]. Therefore, only white bread was included in this study, and other types of bread were excluded.

Coffee: Coffee is a loved beverage consumed by people from all walks of life. On the other hand, it has attracted attention as a unique food in which the four heat-treatment contaminants focused on in this study are formed. Therefore, coffee has been considered in studies evaluating the relationship between heat-treatment pollutants and cancer [75,76,77] and it was divided into three subgroups: instant, ready-to-consume (brewed), and Turkish coffee.

Black tea: Tea is the second most consumed beverage in the world after coffee [78]. Although tea consumption has many health benefits, some studies have reported that tea contains contaminants such as acrylamide, PAHs, and N-nitrosamines that may be harmful to human health [79,80,81]. Turkey has the highest consumption of black tea [82]. In addition, the geography where the study was conducted is the main center of black tea production in Turkey. High consumption of black tea also increases exposure to and health risks from the contaminants found in black tea. Tea is often included in countries with high tea consumption studies examining the relationship between dietary habits and cancer [83,84,85].

Risk scoring: There are different foods in which high levels of each heat-treatment contaminant occur. In addition, the consumption share of each food in daily nutrition also shows significant differences. In light of this information, there is no definite indication that the food groups have a substantial difference from each other in terms of these heat-treatment contaminants. Therefore, the dietary heat-treatment contaminant risk score of each food group was considered the same and defined as 1.

#### 2.3.2. Consumption Frequency

Food consumption frequencies in the “Patient Information and Retrospective 10-Year Nutrition Habits” questionnaire were prepared by considering the information defined in The Japan Public Health Center-JPHC Study and the nutritional habits of individuals living in the geography where the study would be conducted. An increased frequency of consumption of any food group indicates increased exposure to heat-treatment contaminants potentially present in that food. Therefore, as the frequency of consumption increases, the risk score increases. The frequency of food consumption = dietary heat-treatment contaminant risk score is as follows: for the frequency of eating: (never = 1, 1–2 times/year = 2, ≥three times/year = 3, 1–2 times/month = 4, ≥3 times/month = 5, 1–2 times/week = 6, ≥3 times/week = 7, 1–2 times/day = 8 and ≥3 times/day = 9); for the frequency of drinking: (never = 1, 1–2 cups/year = 2, ≥3 cups/year = 3, 1–2 cups/month = 4, ≥3 cups/month = 5, 1–2 cups/week = 6, ≥3 cups/week = 7, 1–2 cups/day = 8 and ≥3 cups/day = 9).

#### 2.3.3. Portion Sizes

In the “Patient Information and Retrospective 10-Year Dietary Habits” questionnaire, the amount of one serving of each food group was determined by using “The Food and Nutrition Photo Catalog: Measurements and Quantities,” prepared by Rakıcıoğlu et al., (2012) [86]. The individuals included in the study were asked to evaluate the previously mentioned images and give three possible answers that would best reflect the portion size of the food solutions in their 10-year dietary habits. The increase in the amount of food consumed from each food group also means that the heat-treatment contaminants in that food are taken more into the body. Therefore, as the portion size consumed increases, the risk score increases. Portion sizes = dietary heat-treatment contaminant risk score was determined as follows: less than half the standard portion size = 1, standard portion size = 2, and more than 1.5-fold = 3.

#### 2.3.4. Cooking Methods

Many cooking methods are used to prepare meat and its products for consumption. Each cooking method directly affects the levels of HCAs, PAHs, acrylamide, and N-nitrosamines in meat and meat products [87,88]. The relationship between cooking methods applied to meat products and cancer has been evaluated in several studies [89,90]. According to the literature, the effect of cooking methods on the level of these four contaminants in meat products varies [91,92,93,94,95]. In this study, the effect of the cooking method on the heat-treatment contaminant level in meat products was considered as deep/pan frying, grilling/barbecue, roasting, baking, and boiling, from the highest to the lowest. In the study, cooking methods for meat products = dietary heat-treatment contaminant risk score was determined as follows: deep/pan frying = 5, grilling/barbecue = 4, roasted = 3, baking = 2, and boiling = 1. The type of oil used in frying, fuel (such as wood or charcoal) used in the grill, and all other parameters, including temperature and time, were excluded.

Several cooking methods are used to prepare potatoes for consumption. Different cooking methods directly affect the levels of heat-treatment contaminants in potatoes [96]. In this study, the cooking methods used to prepare potatoes for consumption were determined to be profound/pan frying and baking. Many researchers have reported that deep/pan frying causes higher levels of heat-treatment contaminant formation in potatoes compared to baking [97,98]. In this study, the cooking methods = dietary heat-treatment contaminant risk score was determined as follows: deep/pan frying = 5 and baking = 2. To compare the risk scores of French fries and meat, the risk scores defined for French fries were accepted as having the same value as similar meat cooking methods. All other parameters were excluded, including the type of oil used for frying, temperature, and time.

Bread is primarily baked in an oven. Since bread is purchased ready-made from the sales point, no further processing of bread at home was considered by consumers. Therefore, the risk score for dietary heat-treatment contaminants in bread was accepted as baking = 2, as in meat and French fries.

In the coffee group, there is no process applied by consumers to make instant, ready-to-consume (brewed), or Turkish coffee ready for consumption. Producers roast coffee beans to make them suitable for consumers. Therefore, the dietary heat-treatment contaminant risk score for coffee products was assumed to be roasting = 3 as in the meat group.

Because consumers brew black tea and prepare it for consumption, there is no cooking method applied to black tea by consumers. Producers dry black tea to make it suitable for consumers. In this study, drying and baking were considered equivalent. Therefore, black tea’s dietary heat-treatment contaminant risk score was accepted as baking (drying) = 2, as in the meat, French fries, and bread groups. All other parameters were not considered, such as different brewing methods for coffee types and black tea, temperature, time, and quantity.

#### 2.3.5. Consumption Mode

The consumption characteristics of each food group vary according to individuals’ dietary habits. Whether meat, French fries, and bread are undercooked, normal, or overcooked affects the levels of heat-treatment contaminants in these foods and dietary exposure [89,99]. In general, as the cooking level of food increases, the level of heat-treatment pollutants in the food also increases. Therefore, the risk scores for meat, French fries, and bread for consumption mode = dietary heat-treatment contaminant were determined as follows: undercooked = 1, normal = 2, and overcooked = 3. The dietary heat-treatment contaminant risk score, depending on the type of coffee included in the study and the consumption of black tea, was accepted as 1.

#### 2.3.6. Dietary Heat-Treatment Contaminant Risk Score Calculation

In this study, according to food types, with each patient having different demographic characteristics, dietary heat-treatment contaminant risk scores were calculated according to the formulas below, according to the retrospective 10-year food consumption characteristics and cancer types of each patient.
Patient risk score = Food group × Frequency of consumption × Portion size × Method of preparation × Method of consumption(1)
Food group risk score = Frequency of consumption × Portion size × Method of preparation × Method of consumption(2)

### 2.4. Data Collection

Data on the dietary habits of the individuals were obtained using a questionnaire prepared for this purpose using the retrospective reminder method. Individuals were asked to recall their dietary habits for the period that started 10 years before they were diagnosed with cancer until the time of the diagnosis. A 10-year eating habit is enough time to have a positive/negative effect on the health of individuals. In addition, Eysteinsdottir et al., (2011) have stated that past diet may be recalled with acceptable accuracy up to 10 years prior, though greater uncertainty exists beyond this period [100]. Considering the health status of the individuals, they were given sufficient time, and interviews were conducted when they felt ready. The researchers applied the relevant questionnaire to the patients face-to-face before they received chemotherapy, and data were collected.

### 2.5. Statistical Analysis

The analyses were completed by transferring the data to the IBM SPSS Statistics 26 and IBM SPSS Amos 23 programs. Before evaluating the study data, validity and reliability analysis was applied to the questionnaire form for food types (meat products, French fries, bread, coffee, and black tea). For the identified food types, four headings were identified: frequency of consumption, portion size, cooking method, and consumption mode. Firstly, corrected-item total correlation coefficients were utilized to determine whether each food type contributes to the scale with these headings. As a result of the item analysis, the item-total correlation coefficients were more significant than +0.25, and no items were removed from the scale. The construct validity and reliability of the scale were examined with confirmatory factor analysis (CFA) and Cronbach’s alpha, respectively. As a result of CFA, it was observed that the fit index values (χ^2^/sd, GFI, AGFI, NFI, NNFI, CFI, RMSEA, and SRMR) of the model provided an acceptable fit and a good fit. Finally, the Cronbach’s alpha (α) internal consistency coefficient of the scale for food types: α = 0.666 for meat products, α = 0.683 for French fries, α = 0.676 for bread, and α = 0.773 for beverages (coffees and black tea) was found to be highly reliable. A value of 0.773 was found to be quite reliable (0.00 < α< 0.40 = scale is not reliable, 0.40 < α < 0.60 = reliable, 0.60 < α < 0.80 = quite reliable, and 0.80 < α < 1.00 = reliable at high degree). In light of all these analyses, it was concluded that the scale is a valid and reliable measurement tool in determining the relationship between the retrospective 10-year dietary habits of patients diagnosed with cancer and the risk score of exposure to dietary heat-treatment contaminants.

When evaluating the data, frequency distributions were provided for categorical variables, and descriptive statistics (mean ± sd, median) were given for numerical variables. In order to decide on the analyses to be applied to the data, the Kolmogorov–Smirnov normality test (*n* > 30) was used for the risk scores. As a result of the trial, it was seen that the total risk score met the assumption of normal distribution. In contrast, the food group risk scores did not meet the assumption of normal distribution, and therefore, both parametric and nonparametric tests were used in the comparisons. An independent Sample T Test was used to determine whether there was a difference in scores between two independent groups, and one-way analysis of variance (ANOVA), the Kruskal–Wallis test, and Friedman analysis were used to determine whether there was a difference in scores among more than two independent groups. The differences between the groups were evaluated using the Tukey and Bonferroni tests. The effects of food groups and food consumption characteristics on cancer types were analyzed using binary logistic regression analysis. Statistically significant differences were observed between the letters in the same group (*p* < 0.05). The logistic regression analysis model prepared by taking the cancer-type-dependent variable is shown in Appendix A.

## 3. Results and Discussion

### 3.1. Demographic Characteristics of the Study Population

The study, which was conducted between 1 January 2020 and 31 October 2022, included 1155 patients with cancer. The sex ratios of the patients who participated in the study were very close to each other, and most of them were married. Approximately 90.0% of the individuals were over 45 years of age. Most of the patients diagnosed with cancer reported having a family history of cancer. The mean body weight and height of the patients were determined as 74.8 ± 14.5 kg (40–154 kg) and 168 ± 9.17 cm (150–195 cm), respectively. Case numbers by cancer type are listed as follows: cancers in the respiratory system > cancers in the urinary system > cancers in the gastrointestinal tract > cancers in the reproductive system > cancers in other systems. The other demographic characteristics of the patients are presented in Table 1.

According to the global cancer data published by the World Cancer Research Fund International in 2020, the top five cancer types are breast, lung, colorectal, prostate, and stomach cancer, respectively [101]. According to the International Agency for Research on Cancer 2020 data, the most common cancer types in Turkey are lung, breast, colorectal, prostate, and thyroid cancer, respectively [102].

### 3.2. Total Risk Scores by Food Groups

The 10-year retrospective nutritional characteristics of some foods that pose a risk for the formation of HCAs, PAHs, acrylamide, and N-nitrosamine before cancer diagnosis and the dietary heat-treatment contaminant risk scores of these foods are shown in Table 2.

According to the average consumption frequency, patients consumed all types of meat and French fries 1–2 times/week and bread 1–2 times/day. The highest average consumption frequency among coffee types was Turkish coffee (1–2 times/week), and the consumption frequency of black tea was determined as 1–2 times/day. The patients consumed all types of meat, French fries, and bread as standard portion sizes on average in their 10-year nutritional history. The average portion size of all brewed coffees, excluding black tea, was approximately one glass/cup. When the average data of the cooking methods were evaluated, it was determined that the patients mostly cooked the meat types by grilling/barbecue and cooked the potatoes by deep/pan frying. The patients consumed different kinds of meat, potatoes, and bread in their normal fried forms.

Within the scope of this study, the total risk scores of dietary heat-treatment contaminants in each food were calculated by considering the nutritional habits of patients diagnosed with cancer. According to full mean total risk scores, the foods were ranked as red meat > white meat > fried potatoes > fish meat > bread > black tea > Turkish coffee > instant coffee > ready-to-drink (brewed) coffee. Statistically significant differences (*p* < 0.05) were found in terms of the total risk scores (median) for heat-treatment contaminants in all food groups. Unlike other food groups, the cooking methods used to prepare meat types and potatoes for consumption are the main reasons for the difference in the heat-treatment contaminant risk scores. Examples of this include data on bread and black tea. Although the frequency of consumption of both these foods is higher than that of meat types and French fries, their average total risk scores are relatively low because they are not prepared for consumption using cooking methods with high levels of heat-treatment contaminants, such as grilling or frying. The fact that the patients consumed small amounts of coffee products affected the low-risk score of this food group. It can also be said that the food consumption mode affects the difference between risk scores. Examples include data on meat types and French fries.

Meat products are a primary source of protein for many consumers [103]. However, high meat consumption has also been linked to obesity, cardiovascular disease, and various types of cancer [104,105,106,107]. The International Agency for Research on Cancer (IARC) has classified meat and processed meat products as possibly carcinogenic to humans (Group 2A) or directly carcinogenic (Group 1) to humans [18]. It has been explained that high levels of HCAs, PAHs, acrylamide, and N-nitrosamines are formed in meat products depending on the raw material properties, the cooking method applied, and the consumption method [19,93,108,109]. This study determined the highest dietary heat-treatment contaminant risk score for the meat group.

French fries have gained popularity as food commodities worldwide [110]. Many people believe that fried food products, such as French fries, are unhealthy [111]. It has been reported that HCAs, PAHs, and acrylamide are formed in French fries, and French fries particularly contribute to dietary acrylamide exposure [31,112,113]. In this study, the heat-treatment contaminant risk score of French fries was found to be high.

Bread, a staple food consumed frequently and in large quantities, is an important component of human nutrition. However, in recent years, concerns have been raised regarding the safety of bread [114]. It is known that HCAs, PAHs, and acrylamide are formed in bread because bread affects people’s health significantly, and therefore, ensuring its safety should be a primary priority [115]. The bread consumption frequency of cancer patients in our study was high.

Coffee and tea are among the most widely consumed beverages in the world. However, many studies have reported that coffee and tea significantly contribute to exposure to heat-treatment contaminants [28,31,85,88,116]. In this study, it was found that patients consumed black tea more frequently and in higher quantities than coffee products.

### 3.3. Total Risk Scores According to the Demographic Characteristics of the Patients

The total risk scores for dietary heat-treatment contaminants according to the demographic characteristics of the patients and statistical comparisons are shown in Table 3.

The mean total risk score from heat-treatment contaminants based on the consumption of meat products, French fries, bread, coffee products, and tea by male patients diagnosed with respiratory, urinary, and other systemic cancers, and all cancer types was significantly higher than that of female patients with the same diagnoses (*p* < 0.05). The frequency of consumption of each food, the cooking methods applied, and the consumption modes of the foods were quite similar in male and female patients. The portion size of the French fries for male patients was the standard portion size, whereas for female patients, it was less than half the standard portion size. Women consume more ready-to-consume brewed coffee and Turkish coffee than men. The highest contribution to the total risk score for heat-treatment contaminants from the consumption of meat products, French fries, bread, coffee products, and tea was from meat products (363 ± 179 and 269 ± 150 for male and female patients, respectively). Men have more muscle tissue than women and therefore require more energy. This may explain the higher dietary heat-treatment contaminant risk score, as men consume more food more frequently. According to the world cancer data published by the World Cancer Research Fund International in 2020, the top five most common cancer types in men are lung cancer, prostate cancer, colorectal cancer, stomach cancer, and liver cancer, and the top five most common cancer types in women are breast cancer, colorectal cancer, lung cancer, cervix uteri cancer, and thyroid cancer, respectively [101]. Grosso et al., (2017) [54] reported that there was no significant difference in unhealthy eating patterns between male and female patients diagnosed with different types of cancer, while Dardzińska et al., (2023) reported that men diagnosed with lung cancer had higher unhealthy eating index scores than women [72].

The mean total risk scores of cancer patients aged 18–44 years and 65+ years due to heat-treatment contaminants based on the consumption of meat products, French fries, bread, coffee products, and tea were close to each other, and the mean total risk score of patients aged 45–64 years was higher than that of patients in different age groups, but this difference was not statistically significant (*p* > 0.05). The mean total risk score from heat-treatment contaminants based on the consumption of meat products, French fries, bread, coffee products, and tea by patients aged 45–64 years diagnosed with cancer types defined in the reproductive and urinary systems was significantly higher than the mean total risk score of patients aged 18–44 years (*p* < 0.05). Patients aged 18–44, 45–64, and 65+ years had similar frequencies of consumption of other types of meat except for fish meat, and the frequency of fish meat consumption among 65+ individuals was higher than that of different age groups. The frequency of French fries, bread, and tea consumption was similar among the age groups. The highest frequency of consumption of all types of coffee was calculated for patients aged 18–44 years. The portion sizes, cooking methods, and consumption modes of each food consumed by individuals aged 18–44, 45–64, and 65+ years were similar. The nutritional characteristics of individuals may vary in different age ranges depending on many factors, such as marital status, work life, and physical and social activities of the same individuals. Marzbani et al., (2019) found that the age range of most of the patients living in Iran and diagnosed with breast cancer was between 40–50 years and there was a positive relationship between breast cancer and unhealthy dietary patterns [117].

The mean total risk score of patients with a history of a smoking habit and with cancers of the respiratory and other systems and any other type of cancer due to heat-treatment contaminants based on the consumption of meat products, French fries, bread, coffee products, and tea was higher than the risk score on non-smoking patients. This difference was statistically significant (*p* < 0.05). The frequency of consumption of meat products, French fries, and bread was similar in smokers and non-smokers; however, the frequency of consumption of coffee products and black tea was higher in smokers than in non-smokers. It is known that smokers often consume other food products, such as tea and coffee, at the same time. Smokers and non-smokers had similar portion sizes, cooking methods, and consumption modes for each food item. Augustsson et al., (1999) reported a strong association between smoking and bladder cancer and a less strong association with kidney cancer but no association with colon and rectal cancer [118]. Gandini et al., (2008) stated that smoking is a known risk factor for many cancers, including lung and colorectal cancer [119]. Dardzińska et al., (2023) explained that smokers are at risk of lung cancer and that smokers and patients diagnosed with lung cancer consume processed meat products more frequently [72].

There was no statistically significant difference between the mean total risk scores of patients diagnosed with cancer and those with or without a family history of cancer due to heat-treatment contaminants based on the consumption of meat products, French fries, bread, coffee products, and tea (*p* > 0.05). The frequency of food consumption, portion sizes, cooking methods, and consumption modes of individuals with and without a family history of cancer were very similar. It has been reported that individuals with a family history of cancer have a higher risk of developing the same cancer due to unhealthy eating habits [120,121,122].

The mean total risk score of obese patients diagnosed with reproductive and urinary cancers and all cancers due to heat-treatment contaminants based on the consumption of meat products, French fries, bread, coffee products, and tea was statistically significantly higher than that of normal-weight patients (*p* < 0.05). Although the frequency of French fries and bread consumption was higher in obese patients than in other patients, the frequency of consumption of other foods was similar. Thin patients were more likely to consume coffee than other individuals. The patients’ food portions increased in direct proportion to their body weight across all food types. The patients’ cooking methods and food consumption modes in the different groups according to body mass index were similar. In this sense, other findings have been reported in the literature. It has been reported that there is no relationship between breast cancer and BMI [117], there is an inverse relationship (premenopausal) [123], and finally, there is a positive relationship [124,125]. Grosso et al., (2017) reported a positive relationship between unhealthy eating patterns and BMI in patients diagnosed with cancer and drew attention to the effect of obesity on cancer risk [126]. Dardzińska et al., (2023) found no significant difference between patients diagnosed with lung cancer with a different body mass index according to non-healthy diet index levels [72].

### 3.4. Risk Scores of the Food Groups According to Cancer Types

In addition, within the scope of the study, the total risk scores of heat-treatment contaminants resulting from the consumption of each food group were calculated according to cancer type and evaluated statistically (Table 4). According to the average total risk scores, the cancer types are listed as follows: cancers in other systems> cancers in the gastrointestinal tract > cancers in the urinary system > cancers in the respiratory system > cancers in the reproductive system. However, no statistically significant difference among the cancer types was found according to the dietary heat-treatment total risk scores (*p* > 0.05). The consumption frequencies of meat products and bread were similar for all the cancer types. The frequency of consumption of French fries in patients diagnosed with cancer types defined in the urinary and reproductive systems is ≥3 times/week, and it is higher (1–2 times/week) than in patients diagnosed with other cancer types. While the frequency of Turkish coffee consumption is higher in patients diagnosed with cancer types defined in the reproductive system than in patients diagnosed with other cancer types, the consumption frequency of other coffee types is similar in patients diagnosed with all cancer types. The frequency of black tea consumption among patients diagnosed with cancer types defined in the lungs and in the gastrointestinal system was higher than in patients diagnosed with other cancer types. The food consumption portions, cooking methods, and consumption modes of patients diagnosed with all cancer types were also similar. There was no statistically significant difference between the heat-treatment contaminant risk scores (median) of the food groups calculated according to the cancer type (*p* > 0.05). The food group that made the highest contribution to the risk score calculated for all cancer types was meat products. Bertuccio et al., (2013) stated that an unhealthy diet model increased the risk of gastric cancer by 50%, Grosso et al., (2017) found a strong association between unhealthy eating patterns and colorectal cancer, Bagheri et al., (2018) and Lozano-Lorca et al., (2022) found a positive relationship between an unhealthy diet and prostate cancer, Hajjar et al., (2022) found that an unhealthy diet increased the risk of kidney cancer, and Dardzińska et al., (2023) reported that patients diagnosed with lung cancer had a high level of unhealthy nutrition index [71,72,126,127,128,129].

### 3.5. The Effect of Food Group and Food Consumption Characteristics on Cancer Types

Only significant results in the logistic regression analysis were included in the study (Table 5).

Model 1A, in which the cancer types defined in the respiratory system were the dependent variable and instant coffee was the independent variable, was statistically significant (*p* < 0.05). According to Model 1A, a one unit increase in instant coffee consumption increases the risk of developing respiratory cancers by 1.034 (Exp (B)). Many studies have reported that increased coffee consumption is associated with lung and laryngeal cancer [130,131,132,133,134]. Galeone et al., (2010) and Al-Dakkak (2011) stated that there is an inverse relationship between coffee consumption and the risk of oral cavity and pharyngeal cancer [84,135].

Model 1C, in which cancer types defined in the respiratory system were the dependent variable and instant coffee portion size was the independent variable, was statistically significant (*p* < 0.05). According to Model 1C, a one unit increase in instant coffee servings increases the risk of developing respiratory cancers by 1.422 (Exp (B)). Tang et al., (2010) reported that high levels of coffee consumption were associated with lung cancer, and Xie et al., (2016) reported a 47.0% increased risk of lung cancer in populations with high levels of coffee consumption compared to those with low levels of coffee consumption [130,132] Some studies have reported that coffee consumption is not associated with lung or laryngeal cancer [136,137].

Model 4B, in which cancer types defined in the urinary system were the dependent variable and the consumption frequency of French fries was the independent variable, was statistically significant (*p* < 0.05). According to Model 4B, a one unit increase in the consumption frequency of French fries increases the risk of developing cancer types defined in the urinary system by 1.151 (Exp (B)). Mucci et al., (2003) and Hogervorst et al., (2008) found no association between dietary (including French fries) acrylamide exposure and prostate, bladder, or kidney cancers [138,139]. According to Seyyed Salehi et al., (2021), N-nitroso compounds found in food and drinking water have no connection to bladder cancer incidence [52]. Red wine, fried eggs, potatoes, and red meat have all been directly linked to an increased risk of bladder cancer, according to De Stefani et al., (2008) [140]. Regular consumption of deep-fried foods, especially French fries, has been reported to increase the risk of prostate cancer [141,142].

Model 3E was statistically significant, in which cancer types defined in the gastrointestinal system were the dependent variable and red meat consumption mode was the independent variable (*p* < 0.05). According to Model 3E, a one unit increase in the consumption mode of red meat increases the risk of developing cancer types defined in the gastrointestinal system by 4.414 (Exp (B)). Consuming meat and fish together, regardless of the cooking method, has been associated with an increased risk of colon cancer [118]. According to Grosso et al., (2017), eating an animal-based diet is strongly associated with a higher risk of colorectal cancer [54]. Martínez Góngora et al., (2019) and Farvid et al., (2021) reported a positive association between meat consumption and colorectal cancer [48,58]. Meat consumption has been linked to cancers of the esophagus, stomach, pancreas, and liver in many studies [126,143,144,145,146,147]. The association between the consumption of well-cooked red meat and colorectal and breast cancers is mostly explained by the formation of carcinogenic HCAs and PAHs [148,149,150].

Model 2F, in which cancer types defined in the reproductive system were the dependent variable and the total consumption frequency was the independent variable, was statistically significant (*p* < 0.05). According to Model 2F, a one unit increase in total consumption frequency increases the risk of developing reproductive cancers by 1.072 (Exp (B)). It has been stated that the increase in the consumption frequency of fried foods, ultra-processed foods, meat, and coffee increases the risk of developing breast, ovarian, and uterine cancers [58,75,117,122,151,152,153,154,155,156]. No correlation was found between cancer type and the cooking methods applied to food. Some studies have reported similar findings [90,157].

## 4. Conclusions

Cancer is a highly complex disease affecting not only one organ but the whole body. Therefore, it is very important to re-evaluate nutrition, as its relationship with cancer has been examined in many studies, from different perspectives. This is because the processes along the food chain and people’s dietary habits cause the formation of toxic compounds in foods, and lifelong dietary exposure occurs with the consumption of these foods. This study considered the dietary habits of patients diagnosed with cancer, such as meat products, French fries, coffee products, bread, and tea, which are considered risky in terms of the formation of HCAs, PAHs, acrylamide, and N-nitrosamines, which are potential carcinogenic compounds in humans. In this study, the relationship between the dietary patterns of the patients for 10 years before cancer diagnosis and cancer types was evaluated according to the dietary heat-treatment contaminant risk score. Many results found in this study support those of previous studies examining the relationship between nutrition and cancer. To explain the relationship between nutrition and cancer, studies involving more foods and different dietary patterns should be conducted in large populations.

In addition, oncology nurses who care for patients diagnosed with cancer play a role in providing individualized nutrition and guidance. In this sense, oncology nurses should systematically monitor and evaluate the nutritional status of all patients with cancer, especially those who maintain risky eating habits.

This study was carried out on some foods consumed by patients admitted to an oncology center in Turkey in a 10-year period retrospectively and some heat-treatment contaminants in these foods. The heat-treatment contaminant levels in the foods in question depend on the type of raw material, the production method, etc., and may vary depending on factors. In addition, it should not be forgotten that the rate of consumption of these foods by patients may differ according to geography and culture. Therefore, the obtained results cannot be generalized to other patients diagnosed with similar cancer.

## Figures and Tables

**Table 1 foods-12-02320-t001:** Demographic characteristics of patients and cancer types.

	Number of Persons (*n* = 1155)	Percentage (%)
Sex		
Male	579	50.1
Female	576	49.9
Marital status		
Married	1029	89.1
Single	126	10.9
Age groups		
18–44	142	12.3
45–64	416	36.0
65+	597	51.7
Profession		
Housewife	420	36.3
Public employee	98	8.50
Retired	472	40.9
Student	8	0.70
Private sector employee	157	13.6
Cigarette consumption		
Yes	561	48.6
No	594	51.4
Family history of cancer		
Yes	784	67.9
No	371	32.1
Body mass index (kg/m^2^)		
Underweight (<18.5)	23	2.00
Normal weight (18.5–24.9)	430	37.2
Overweight (25–29.9)	494	42.8
Obese (≥30)	208	18.0
Type of cancer		
Cancers in the reproductive system ^a^	126	10.9
Cancers in the gastrointestinal tract ^b^	254	22.0
Cancers in the urinary system ^c^	298	25.8
Cancers in the respiratory system ^d^	334	28.9
Cancers in other systems ^e^	143	12.4

^a^ Reproductive system cancer types = breast, uterus, and ovary. ^b^ Gastrointestinal system cancer types = esophagus, stomach, colorectum, pancreas, and liver. ^c^ Urinary system cancer types = prostate, bladder, and kidney. ^d^ Respiratory cancer types = lung, larynx, and oral cavity/pharyngeal. ^e^ Other system cancer types = brain, thyroid, lymphatic malignancies, skin, oro- and hypopharynx, and hematology.

**Table 2 foods-12-02320-t002:** Consumption characteristics and risk scores of foods.

Food Groups	Consumption Frequency		Portion Amount		Cooking Method		Consumption Mode		Total Risk Score	
	Mean ± SD	Median (Min.–Max.)	Mean ± SD	Median (Min.–Max.)	Mean ± SD	Median (Min.–Max.)	Mean ± SD	Median (Min.–Max.)	Mean ± SD	Median (Min.–Max.)
Meat (red) *	6.24 ± 1.25	7 (1–8)	1.94 ± 0.78	2 (1–3)	4.24 ± 1.28	5 (1–6)	2.07 ± 0.51	2 (1–3)	112 ± 66.4	112 (1–525) ^a^
Meat (white) *	6.24 ± 1.25	7 (1–9)	1.91 ± 0.79	2 (1–3)	4.11 ± 1.27	5 (1–5)	2.07 ± 0.48	2 (1–3)	108 ± 66.5	100 (1–525) ^a^
Meat (fish) *	5.93 ± 1.35	6 (1–9)	1.93 ± 0.79	2 (1–3)	3.85 ± 1.62	5 (1–5)	2.09 ± 0.53	2 (1–3)	95.5 ± 66.2	84 (1–450) ^a^
French fries	6.48 ± 1.80	7 (1–9)	1.72 ± 0.78	2 (1–3)	4.66 ± 1.47	5 (1–7)	2.03 ± 0.51	2 (1–3)	107 ± 71.9	90 (2–420) ^b^
Bread	7.96 ± 1.87	9 (1–9)	1.66 ± 0.80	1 (1–3)	2.00 ± 0.00	2 (2–2)	1.87 ± 0.35	2 (1–3)	50.5 ± 22.4	18 (2–240) ^d^
Coffee (instant) *	1.65 ± 1.85	1 (1–9)	1.07 ± 0.52	1 (1–3)	3.00 ± 0.00	3 (3–3)	1.00 ± 0.00	1 (1–1)	5.36 ± 5.73	3 (3–60) ^e^
Coffee (ready to drink) *	1.15 ± 0.98	1 (1–9)	1.03 ± 0.35	1 (1–3)	3.00 ± 0.00	3 (3–3)	1.00 ± 0.00	1 (1–1)	3.77 ± 4.06	3 (3–60) ^e^
Coffee (Turkish coffee) *	5.86 ± 3.03	1 (1–9)	1.28 ± 1.01	1 (1–3)	3.00 ± 0.00	3 (3–3)	1.00 ± 0.00	1 (1–1)	22.8 ± 10.5	5 (3–120) ^e^
Black tea	7.72 ± 3.87	9 (1–9)	2.76 ± 1.98	1 (1–3)	2.00 ± 0.00	2 (2–2)	1.00 ± 0.00	1 (1–1)	46.8 ± 23.1	28 (2–450) ^c^

SD = standard deviation, Min. = minimum, and Max = maximum. * The heat-treatment contaminant total risk scores of foods in the meat and coffee groups were combined. Different letters in the same group indicate statistically significant differences (*p* < 0.05).

**Table 3 foods-12-02320-t003:** Risk scores according to the demographic characteristics of the patients.

Demographic Features	Total Risk Score	Types of Cancer				
		Cancers in the Reproductive System ^a^	Cancers in the Gastrointestinal Tract ^b^	Cancers in the Urinary System ^c^	Cancers in the Respiratory System ^d^	Cancers in Other Systems ^e^
	Mean ± SD	Mean ± SD	Mean ± SD	Mean ± SD	Mean ± SD	Mean ± SD
Sex						
Male	584 ± 207 ^x^	Not calculated	553 ± 219 ^x^	611 ± 200 ^x^	575 ± 196 ^x^	622 ± 218 ^x^
Female	462 ± 185 ^y^	496 ± 182	487 ± 209 ^x^	461 ± 185 ^y^	391 ± 153 ^y^	407 ± 171 ^y^
Age groups						
18–44	515 ± 167 ^x^	425 ± 117 ^y^	544 ± 21 ^x^	448 ± 128 ^y^	559 ± 135 ^x^	604 ± 200 ^x^
45–64	546 ± 204 ^x^	552 ± 182 ^x^	523 ± 246 ^x^	596 ± 184 ^x^	518 ± 177 ^x^	555 ± 235 ^x^
65+	510 ± 213 ^x^	474 ± 194 ^x,y^	532 ± 195 ^x^	502 ± 213 ^x,y^	515 ± 233 ^x^	520 ± 231 ^x^
Cigarette consumption						
Yes	550 ± 191 ^x^	505 ± 171 ^x^	534 ± 172 ^x^	535 ± 186 ^x^	553 ± 196 ^x^	612 ± 217 ^x^
No	497 ± 215 ^y^	492 ± 188 ^x^	526 ± 247 ^x^	522 ± 217 ^x^	446 ± 198 ^y^	415 ± 184 ^y^
Family history of cancer						
Yes	534 ± 204 ^x^	518 ± 176 ^x^	538 ± 227 ^x^	521 ± 196 ^x^	542 ± 204 ^x^	565 ± 223 ^x^
No	499 ± 208 ^x^	432 ± 187 ^x^	517 ± 202 ^x^	543 ± 229 ^x^	476 ± 192 ^x^	501 ± 232 ^x^
Body mass index (kg/m^2^)						
Underweight (<18.5)	490 ± 193 ^x,y^	515 ± 170 ^x,y^	530 ± 126 ^x^	425 ± 121 ^x,y^	498 ± 160 ^x^	417 ± 155 ^x^
Normal weight (18.5–24.9)	485 ± 196 ^y^	412 ± 146 ^y^	510 ± 225 ^x^	466 ± 161 ^y^	493 ± 195 ^x^	511 ± 213 ^x^
Overweight (25–29.9)	535 ± 207 ^x,y^	492 ± 171 ^x,y^	539 ± 224 ^x^	535 ± 211 ^x,y^	536 ± 197 ^x^	611 ± 253 ^x^
Obese (≥30)	575 ± 211 ^x^	584 ± 200 ^x^	561 ± 182 ^x^	616 ± 230 ^x^	556 ± 243 ^x^	506 ± 190 ^x^

^a^ Reproductive system cancer types = breast, uterus, and ovary. ^b^ Gastrointestinal system cancer types = esophagus, stomach, colorectum, pancreas, and liver. ^c^ Urinary system cancer types = prostate, bladder, and kidney. ^d^ Respiratory cancer types = lung, larynx, and oral cavity/pharyngeal. ^e^ Other system cancer types = brain, thyroid, lymphatic malignancies, skin, oro- and hypopharynx, and hematology. ^x,y^ Different letters in the same group indicate statistically significant differences (*p* < 0.05).

**Table 4 foods-12-02320-t004:** Total risk scores of cancer types by food groups.

Types of Cancer	Total Risk Score	Meat (Red)	Meat (White)	Meat (Fish)	French Fries	Bread	Coffee (İnstant)	Coffee (Ready to Drink)	Coffee (Turkish Coffee)	Black Tea
	Mean ± SD	Median (Min–Max)	Median (Min–Max)	Median (Min–Max)	Median (Min–Max)	Median (Min–Max)	Median (Min–Max)	Median (Min–Max)	Median (Min–Max)	Median (Min–Max)
Cancers in the reproductive system ^a^	496 ± 182 ^x^	105 (1–210) ^x^	84 (1–210) ^x^	70 (1–210) ^x^	90 (2–315) ^x^	18 (6–240) ^x^	3 (3–18) ^x^	3 (3–2) ^x^	6 (3–120) ^x^	9 (2–90) ^x^
Cancers in the gastrointestinal tract ^b^	530 ± 217 ^x^	120 (1–525) ^x^	112 (1–525) ^x^	100 (1–450) ^x^	90 (2–315) ^x^	18 (2–54) ^x^	3 (3–30) ^x^	3 (3–60) ^x^	3 (3–45) ^x^	18 (2–90) ^x^
Cancers in the urinary system ^c^	527 ± 205 ^x^	100 (1–315) ^x^	96 (1–315) ^x^	84 (1–375) ^x^	90 (2–420) ^x^	18 (2–160) ^x^	3 (3–50) ^x^	3 (3–45) ^x^	3 (3–72) ^x^	18 (2–450) ^x^
Cancers in the respiratory system ^d^	520 ± 202 ^x^	112 (1–270) ^x^	96 (1–315) ^x^	75 (1–252) ^x^	84 (2–360) ^x^	18 (2–180) ^x^	3 (3–60) ^x^	3 (3–35) ^x^	3 (1–45) ^x^	36 (2–135) ^x^
Cancers in other systems ^e^	544 ± 226 ^x^	120 (1–270) ^x^	120 (1–270) ^x^	96 (1–315) ^x^	72 (2–315) ^x^	18 (2–160) ^x^	3 (3–18) ^x^	3 (3–45) ^x^	3 (3–45) ^x^	9 (2–90) ^x^

^a^ Reproductive system cancer types = breast, uterus, and ovary. ^b^ Gastrointestinal system cancer types = esophagus, stomach, colorectum, pancreas, and liver. ^c^ Urinary system cancer types = prostate, bladder, and kidney. ^d^ Respiratory cancer types = lung, larynx, and oral cavity/pharyngeal. ^e^ Other system cancer types = brain, thyroid, lymphatic malignancies, skin, oro- and hypopharynx, and hematology. ^x^ Different letters in the same group indicate statistically significant differences (*p* < 0.05).

**Table 5 foods-12-02320-t005:** Regression analysis results.

Models	Independent Variables	B	SE	Wald	df	Exp (B)	95% C.I. for Exp (B)		Model Summary
							Lower	Upper	
Model 1A	Coffee (instant)	0.033	0.015	4.573	1	1.034	1.003	1.065	x^2^ = 4.572 *p* = 0.032 CCR = %75.3
	Constant	−1.190	0.107	123.472	1	0.304			
	Portion amount								
Model 1C	Coffee (instant)	0.352	0.166	4.504	1	1.422	1.027	1.969	x^2^ = 4.430 *p* = 0.035 CCR = %75.3
	Constant	−1.490	0.207	51.740	1	0.225			
	Consumption frequency								
Model 4B	French fries	0.140	0.062	5.117	1	1.151	1.019	1.299	x^2^ = 5.678 *p* = 0.017 CCR = %74.2
	Constant	−1.983	0.427	21.555	1	0.138			
	Consumption mode								
Model 3E	Meat (red)	1.485	0.615	5.836	1	4.414	1.323	14.722	x^2^ = 5.619 *p* = 0.018 CCR = %78.0
	Constant	−2.787	0.641	18.906	1	0.062			
Model 2F	Total consumption frequency	0.069	0.021	10.359	1	1.072	1.027	1.118	x^2^= 13.484 *p* = 0.001 CCR = %83.5
	Constant	−3.823	1.152	11.016	1	0.022			

*p* = significance level (*p* < 0.05), B = regression coefficient, Exp (B) = odds ratio, CI = confidence interval, SE = standard error, dF = degrees of freedom, and CCR = correct classification rate.

## Data Availability

The data used to support the findings of this study can be made available by the corresponding author upon request.

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
