# Peer review of "Dietary Heat-Treatment Contaminants Exposure and Cancer: A Case Study from Turkey"

_foods, 2023, doi:10.3390/foods12122320_

Round 1
Reviewer 1 Report
The presented study aimed to evaluate risk between dietary heat-treatment contaminants exposure and cancer development among Türkiye population. Topic is important, although many papers related to the establishment of connection among the intake of heat-treatment contaminants and the cancer development are already recorded by other authors. Novelty is to some extent questionable. However, the paper complements previous research and provides to some extent a broader insight into risk from thermal processing contaminants.
English language of MS should be revised and corrected.
The manuscript is partially incomprehensible.
Minor grammatical errors are also present, as in Line 345: “the most common cancer types in TürkiyeRe are lung”
Some statements need to be more clearly presented, as in already in Abstract: “According to cancer types the highest and lowest dietary heat-treatment contaminant risk scores were determined as cancer types defined in other systems and cancer types defined in reproductive system, respectively.”(Line 17-20) or Line 236-238: “Many researchers have reported higher levels of heat-treatment contaminant formation in potatoes than in deep/pan frying.”
Materials and Methods
Important!
Please clarify how it was estimated or calculated: Total risk scores according to demographic characteristics of patients and Total risk scores of cancer types by food groups
Results and discussion
Table 3. Is the Portion amount (Table 3) equal to Portion sizes (Materials and Methods)? In Material and Methods section state: “Portion sizes=Dietary heat-treatment contaminant Risk score is as follows: less than half the standard portion size=1, standard portion size=2, and more than 1.5-fold=3.” - so obviously there are three options. Please explain how there are four or six options for the same in the table 3: e.g. French fries: 2 (1−6) or Meat (red): 2 (1−4). The same question is for Consumption frequency for Meat (red): 7 (1−8) as in Material and Methods section state 9 options.
The discussion needs to be refined and clarified and could be shortened. The listed literature sources are relevant and comprehensive.
Conclusion
Some specific guidelines should be provided.
Please reformulate: “Systematic follow-up and training of nurses, physicians, dietitians, and all other health professionals on current information on nutrition and cancer, including heat treatment contaminants and foods specific to this study, may help to reduce the number of cancer diagnoses due to heat treatment contaminants today and in the future.”
References
Some citations within the text of the manuscript are missing in the reference list: e.g. Ifegwu and Anyakora, 2015
Reference should be thoroughly checked and corrected.
The presented study aimed to evaluate risk between dietary heat-treatment contaminants exposure and cancer development among Türkiye population. Topic is important, although many papers related to the establishment of connection among the intake of heat-treatment contaminants and the cancer development are already recorded by other authors. Novelty is to some extent questionable. However, the paper complements previous research and provides to some extent a broader insight into risk from thermal processing contaminants.
English language of MS should be revised and corrected.
The manuscript is partially incomprehensible.
Minor grammatical errors are also present, as in Line 345: “the most common cancer types in TürkiyeRe are lung”
Some statements need to be more clearly presented, as in already in Abstract: “According to cancer types the highest and lowest dietary heat-treatment contaminant risk scores were determined as cancer types defined in other systems and cancer types defined in reproductive system, respectively.”(Line 17-20) or Line 236-238: “Many researchers have reported higher levels of heat-treatment contaminant formation in potatoes than in deep/pan frying.”
Materials and Methods
Important!
Please clarify how it was estimated or calculated: Total risk scores according to demographic characteristics of patients and Total risk scores of cancer types by food groups
Results and discussion
Table 3. Is the Portion amount (Table 3) equal to Portion sizes (Materials and Methods)? In Material and Methods section state: “Portion sizes=Dietary heat-treatment contaminant Risk score is as follows: less than half the standard portion size=1, standard portion size=2, and more than 1.5-fold=3.” - so obviously there are three options. Please explain how there are four or six options for the same in the table 3: e.g. French fries: 2 (1−6) or Meat (red): 2 (1−4). The same question is for Consumption frequency for Meat (red): 7 (1−8) as in Material and Methods section state 9 options.
The discussion needs to be refined and clarified and could be shortened. The listed literature sources are relevant and comprehensive.
Conclusion
Some specific guidelines should be provided.
Please reformulate: “Systematic follow-up and training of nurses, physicians, dietitians, and all other health professionals on current information on nutrition and cancer, including heat treatment contaminants and foods specific to this study, may help to reduce the number of cancer diagnoses due to heat treatment contaminants today and in the future.”
References
Some citations within the text of the manuscript are missing in the reference list: e.g. Ifegwu and Anyakora, 2015
Reference should be thoroughly checked and corrected.
Reviewer 2 Report
The study outlay is interesting, methods okey, language okey.
Very complex research area indeed. The results of such queries depend on the interviewer, the protocol, the quality of input by the study objects. Was the protocol tested before the study, eg by repeated tests on the same individuals - testing for conformal answers?
The study is interesting due to turkish coffee ( is this much stronger and more heated than other kinds?) and the large intake of white bread (what about the french baguettes?). I was not aware of this before. Red meat, grilling and overcoking food, ready made industrialised food are known risk factors, but fish??? Salmon and other types of fat fish - I believed was the most healthy choice - being a home cook I never overcook fish - so is overcooked fish the "natural" in your country.
Some of the tables are complex and do not contribute to the main messages in this study, they can be transferred to a supplement.
The results and discusion could be condensed and stressing new and divergent results from othert studies.
A limitation paragraph should be inserted at the end.
BMI is not the whole thing, extremely fat with BMI>40 is even more interesting. Visceral fat high and its relation to total body fat (even higher in fragile patients) is also interesting.
Hot spiced food is a problem in esophageal and gastric cancers. So is intake of alcoholic beverages.
.
Reviewer 3 Report
In this study, the author presents a case study of dietary heat-treatment contaminants exposure in relation to cancer. The study examined the contamination risk scores of various foods based on dietary heat treatment, finding red meat and ready-to-drink coffee to have the highest and lowest scores, respectively. The contamination risk scores varied significantly based on cancer patients' demographic characteristics. Cancer types defined in other systems had the highest contamination risk scores, while cancer types defined in the reproductive system had the lowest scores. The study also revealed associations between specific dietary habits (such as instant coffee consumption, frequent French fries consumption, and meat product consumption) and respiratory, urinary, and gastrointestinal system cancer types, respectively. Overall, this study provides valuable insights into the relationship between dietary habits and cancer, serving as a potential resource for future research in this area.
The expression needs to be improved before published.
